# Sandwich Fluorescence Detection of Foodborne Pathogen *Staphylococcus aureus* with CD Fluorescence Signal Amplification in Food Samples

**DOI:** 10.3390/foods11070945

**Published:** 2022-03-25

**Authors:** Han Du, Tao Ping, Wei Wu, Qingli Yang

**Affiliations:** 1College of Food Science and Engineering, Shandong Agricultural University, Taian 271018, China; 2019010035@sdau.edu.cn; 2College of Food Science and Engineering, Qingdao Agricultural University, Qingdao 266109, China; pingtao@qau.edu.cn (T.P.); wuweiouc@126.com (W.W.); 3Qingdao Institute of Special Food, Qingdao Agricultural University, No. 700 Changcheng Road, Qingdao 266109, China

**Keywords:** fluorescence biosensor, carbon dot, aptamer, *Staphylococcus aureus*

## Abstract

Timely detection of *Staphylococcus aureus* (*S. aureus*) is critical because it can multiply to disease−causing levels in a matter of hours. Herein, a simple and sensitive DNA tetrahedral (Td) fluorescence signal amplifier with blue carbon quantum dots (bCDs) was prepared for sandwich detection of *S. aureus*. bCD was modified at the apex of Td, and an aptamer on Td was used to accurately identify and “adsorb” the amplifier to the surface of *S. aureus*. Atomic force microscopy (AFM) demonstrates the successful preparation of this signal amplifier. The fluorescence intensity emitted in this strategy increased 4.72 times. The strategy showed a stronger fluorescence intensity change, sensitivity (linear range of 7.22 × 10^0^–1.44  × 10^9^ CFU/mL with a LOD of 4 CFU/mL), and selectivity. The recovery rate in qualified pasteurized milk and drinking water samples was 96.54% to 104.72%. Compared with simple aptamer sandwich detection, these fluorescence signal amplifiers have improved fluorescence detection of *S. aureus*. Additionally, this fluorescent signal amplification strategy may be applied to the detection of other food pathogens or environmental microorganisms in the future.

## 1. Introduction

*Staphylococcus aureus* (*S. aureus*) causes foodborne intoxication such as pneumonia, pseudomembranous colitis, pericarditis, and even sepsis, which can also cause a wide range of infections in the healthcare sector [1]. Staphylococcal food poisoning (SFP) is derived from a variety of food types [2]. Milk and dairy products, especially cheese, often produce toxic levels of *S. aureus* [3]. *S. aureus* may be present in samples of food contact surfaces associated with their hosts, such as refrigerators, food bags, even shelves [4]. Moreover, *S. aureus* usually starts out in small numbers, infects food or humans, and quickly proliferate to disease-causing numbers in 8–10 h. Therefore, detection of *S. aureus* contamination in time is crucial.

Many methods have been developed for *S. aureus* detection over the past few decades. For example, colony counting is the most accurate method, and its purpose is to accurately quantify even a small number of *S. aureus*, while time consumption, professional operation scenes, and personnel are inevitable defects of this method. Moreover, nucleic acid-based polymerase chain reaction (PCR) [5] and loop-mediated isothermal amplification (LAMP) [6] have been innovated to overcome the time consumption, but they are constrained by the demands of specialized personnel and sophisticated equipment. In addition, the enzyme-linked immunosorbent assay (ELISA) [7] method based on the immune function of antigen-antibody was developed to eliminate false positives while cumbersome sample processing and strict operating conditions limit it. There are other emerging technologies, such as biosensing technology, that can convert the concentration signal of the target into a fluorescent signal that comes into view. Fluorescence sensing technology has the advantages of fast analysis speed, selectivity, sensitivity, being easy to carry and field analysis, miniaturization, and multi-function [8,9], and has become a hotspot of current research [10].

The selection of fluorescent signal molecules is very important for detection. When analyzing food and living things, the safety and biocompatibility of fluorescent signal molecules have a significant impact on whether new contaminants will be introduced into the analysis object [11]. Carbon quantum dots (CDs) with low toxicity even non−toxicity are safer and more reliable and have been employed as fluorescent signal molecules in fluorescence detection, photocatalysis, and biosensors [12,13]. CDs are relatively easy to synthesize and possess excellent biocompatibility [14], favorable chemical stability [15], and high photostability [16,17]. For the detection of foodborne pathogens, amplification of weak concentration signals can effectively improve the detection performance. Theoretically, when the number of fluorescent signal molecules aggregates on a single target increases, the fluorescence signal it can express becomes stronger [18]. For example, Rhodamine B (RhB), Acrylamide, and N, N’-methylenebisacrylamide were prepared into P(AM-BA-RhB) fluorescent polymer microspheres by parallel synthesis [19]. Han et al. found that CdSe/ZnSe/ZnS quantum dots (QDs) nanobeads can effectively amplify the signal of hepatitis B surface antigen proteins [20]. In another work, *S. aureus* was used as a “pocket” to carry a lot of CDs; the antibodies on its surface helped it find the target and anchored to its surface to glow green [21]. This study also indicated that the degree to which the pockets of *S. aureus* cells were inactivated affects whether there were new contaminants in the sample. Therefore, a more convenient and efficient fluorescence signal amplification strategy needs to be established.

Magnetic materials remain stable in complex non-magnetic samples and can chemically modify functional molecules on their surfaces, expanding their applications [22,23]. Previous studies have combined Fe_3_O_4_-based magnetic composite materials with fluorescence or colorimetry to easily separate and enrich targets using external magnetic fields [24,25], further improving the sensitivity of optical biosensors, suitable for the detection of trace substances in complex samples [26,27]. Aptamers are functional single-stranded nucleic acids obtained through multiple rounds of screening [28,29]. The binding specificity of aptamer and target is high and stable, and each target can bind thousands of aptamers on its surface [30]. However, aptamers may intertwine with each other, affecting their recognition of the target [31,32]. The artificial DNA three-dimensional structure inspires to solve this problem. DNA tetrahedron (Td) formed by DNA self-assembly through base complementary pairing is a candidate [33,34]. The stable triangular structure can not only anchor the aptamers firmly but also distance the aptamers from each other, preventing them from becoming too close [35,36].

In the current study, an *S. aureus* fluorescent signal amplification strategy was proposed by using one vertex of DNA tetrahedron as an anchor for aptamer and the other three vertices for CDs. Another *S. aureus* aptamer was prepared to form a DNA tetrahedron−aptamer (Td−apt) structure on the DNA tetrahedron, which was modified on the surface of magnetic beads (MB) for separation and enrichment of the target.

## 2. Materials and Methods

### 2.1. Materials and Apparatus

All the single-stranded oligonucleotides of the Td listed in Appendix A were synthesized by Sangon Biotech Co. (Shanghai, China). The Luria-Bertani culture (LB, pH = 7.4) was bought from Sinopharm (Beijing, China), HEPES (pH = 7.4), NaCl, 1-ethyl-3-(3-dimethylaminopropyl) carbodiimide hydrochloride (EDC) was bought from Sigma-Aldrich LLC. (Shanghai, China). Biotin, quinine sulfate, N-hydroxysulfosuccinimide (NHS) were provided by Tianjin Macklin (Shanghai, China). streptavidin-functionalized magnetic beads (SA−MBs) with a diameter of 2–3 μm were purchased from BaseLine ChromTech Research Centre (Tianjin, China). DNA loading buffer (6×) and DNA Markers were provided by Sangon Biotechnology Co., Ltd. (Shanghai, China). All solutions with a resistivity of 18.2 MΩ cm were prepared using a Milli-Q water purification system (Sartorius, Germany).

### 2.2. Strains Cultivation and Cytometry

In this work, *S. aureus* (ATCC 6538) was the target, *Escherichia coli* O157:H7 (*E. coli* O157:H7, ATCC 8739), *Salmonella enterica serovar Typhimurium* (*S. Typhimurium*, ATCC 14028), Methicillin−resistant *Staphylococcus aureus* (*MRSA*, ATCC 43300), and *Vibrio parahaemolyticus* (*V. parahaemolyticus*, ATCC 17802) were used as interference items to test the specificity. They were provided by Guangdong Microbial Culture Collection Center (Guangzhou, China). In order to prepare a working suspension of the above strains as in previous work [37], the five bacterial strains were incubated overnight in 10 mL LB medium at 37 °C, separately. Take 1 mL bacteria liquid to centrifuge for 8 min at 5000 rpm and wash with equal volume HEPES buffer 3 times, and bacteria at a concentration of 10^1^–10^8^ CFU/mL were obtained by continuous 10-fold gradient dilution with sterile HEPES buffer. Then the bacteria cells were resuspended in 1 mL HEPES buffer. For colony count, drop 10 μL of suspended liquid onto the blood count plate. Samples of bacteria were counted with a microscope.

### 2.3. Synthesis of CD

The synthesis of CD was adapted from previous reports [38]. A total of 0.399 g of urea and 1.065 g of citric acid were dissolved in 20 mL of ultrapure water and mixed evenly. The mixed liquid was transferred to a high-pressure reactor for water and heated to 160 °C for the hydration reaction. After 6 h, the reaction liquid was cooled to room temperature. The resulting solution was dialyzed in a 3500 Da dialysis bag for 4 h, and the solution outside the dialysis bag was collected. Finally, these solutions were dialyzed in a 500 Da dialysis bag to remove unreacted small molecules and to obtain pure blue CD (bCD). The prepared bCD was stored at 4 °C away from light for later use. The detailed characterizations were described in the Appendix A.

### 2.4. Preparation of Td

In this work, we designed two Td constructures (Td1 and Td2) to couple with SA−MB and bCD, respectively. To prepare Td with high yield, the authors referred to and modified the previous report [39,40]. The four single strands of Td (T1, T2−des, T3−des, and T4−des) were mixed with the same concentration (4 μM) and volume. After the chain disconnecting process at 95 °C for 10 min, the mixture was cooled to 4 °C for at least 30 min. After that, the high yield Td1 could be obtained by mixing Apt1 (4 μM) with the above solution in a ratio of 1:4 by volume and standing. The Td2 solution was prepared in the same way, except that Apt1, T2−des, T3−des, and T4−des in the above operation was replaced by Apt2, T2−NH_2_, T3−NH_2_, and T4−NH_2_. The detailed characterizations and ratio optimization of raw materials were described in the Appendix A.

### 2.5. Preparation of Td1 Modified MBs (Td1−MBs)

Td1−MBs were produced by the strong interaction between desthiobiotin and SA. Suck up 300 μL of SA−MB and rinse it three times with HEPES buffer. A total of 100 μL of Td1 and HEPES re-suspended SA−MB were evenly mixed and gently overturned and oscillated at 37 °C for 30 min. After magnetic separation, the supernatant was discarded and the precipitate was suspended in HEPES again, leading to the formation of Td1−MBs.

### 2.6. Preparation of Td2−bCD

Firstly, 2 μL bCD and 100 μL Td2 were mixed evenly, followed by freshly configured 7.5 mL of 10 mg/mL EDC and 10.13 mL of 10 mg/mL NHS. When activated by EDC, abundant −COOH on bCD surface will react with −NH_2_ modified on Td2 to form semi-stable NHS; the semi-stable NHS will further react with a −NH_2_ to form an amide link. After sufficient vortices, the mixed solution was kept at room temperature overnight, away from light.

### 2.7. Sandwich Detection of S. aureus with CD Fluorescence Signal Amplification

#### 2.7.1. Principle

The principle is shown in Figure 1. The targets were captured and enriched based on magnetic separation. Aptamers on Td1−MBs can precisely pull *S. aureus* out of its complex matrix, laying a foundation for the subsequent release of fluorescence signals. Td2−bCD acts as another target catcher and signal amplifier. The magnetic bead enriched *S. aureus* attracted abundant Apt2 and covered its surface with bCD, leading to the formation of Td1−MB@*S. aureus*@Td2−Bcd complex. In order to release the fluorescence of bCD, a competitive reaction of SA with biotin and desthiobiotin was introduced. Taking advantage of the fact that biotin has hundreds of times higher affinity for SA than desthiobiotin [41], Td1@*S. aureus*@Td2−bCD bound to the magnetic beads was detached and dispersed in the buffer added after magnetic separation. With enough MBs and CD, the more *S. aureus* was caught, the stronger the fluorescence signal will be detected in the final buffer.

#### 2.7.2. Feasibility Verification and Signal Amplification Confirmation

50 μL of 10^8^ CFU/mL *S. aureus* was dropped in 300 µL Td1−MB HEPES solution, and the mixture was incubated at 37 °C for 45 min. The first magnetic separation was then performed to remove the non-target substance. A total of 200 μL Td2−bCD was added to the precipitate, and then the precipitate was mixed by vortex and gently oscillated for 30 min under dark conditions. The second magnetic separation was performed at this time to remove excess Td2−bCD. After cleaning, add 1 mM biotin solution to the precipitation, mix well and gently shake for 30 min. Finally, the supernatant after magnetic separation of the solution was taken to measure fluorescence at 440 nm and compared with the control group without a target.

In order to confirm signal amplification, another two tests proceeded. In one test, the Td2−bCD was replaced by Apt2−bCD, and in the other one, the Td1−MB and Td2−bCD were replaced by Apt1−MB and Apt2−bCD.

#### 2.7.3. Detection of the Concentration of *S. aureus* and Specificity Test

Different concentrations of *S. aureus* (10−fold the concentration gradient) suspensions were dropped in 300 μL Td1−MB HEPES solution, separately. The following steps were the same as Section 2.7.2, and the detected fluorescence spectrums were plotted in a diagram.

For specificity test, *S. aureus*, *E. coli* O157:H7, *S. Typhimurium*, MRSA, *V. parahaemolyticus*, and the same concentration mixture of the above strains (10^3^ CFU/mL) were detected by this proposed strategy in Section 2.7.2, and the detected fluorescence spectrums were plotted in a diagram.

#### 2.7.4. Detection Assays in Practical Samples

The recovery experiment has proceeded in broth, qualified pasteurized milk, and drinking water samples. The concentration of *S. aureus* detected by this method was compared with the concentration added, and the recovery rate was calculated. The error bar in the result was calculated by 3 repetitions.

## 3. Results and Discussion

### 3.1. Synthesis and Characterization of bCD

N-doping CDs were a promising doping system [42,43]. Here, an n-doped citric acid precursor bCD was prepared by using the amino group in urea as a nitrogen source. Transmission electron microscopy (TEM), high-resolution transmission electron microscopy (HRTEM), and atomic force microscopy (AFM) of bCD were performed to confirm its morphological characteristics and the lattice spacing of the bCD was further examined by HRTEM (Figure 1). As shown in Figure 1A Inset, the lattice spacing was 0.21 nm, which corresponds to the (100) in−plane lattice of graphene. In addition, Figure 1A,C indicated that the prepared bCD has a uniform size and high monodispersity. Moreover, the bCD had taken breathing intervals from 1.5 to 3.5 nm with an average particle size of (2.37 ± 0.39) nm.

Furthermore, the surface structure and functional group composition of bCDs were studied by Fourier Transform Infrared (FT−−IR). As illustrated in Figure 1D, the absorption peak between 3400 cm^−1^ and 3600 cm^−1^ was caused by the stretching vibration of O−H. The absorption peaks at 1613 cm^−1^ and 1590 cm^−1^ were attributed to C = O and C = C stretching vibration. These results suggested that the surface of the blue carbon spot had a highly hydrophilic carboxyl ligand on it.

In order to explore the luminescence properties of bCDs, the relative quantum fluorescence yield, UV/Vis spectrum, and fluorescence spectrum of bCDs were analyzed. The quantum fluorescence yield of bCDs was calculated by using quinine sulfate as a fluorescence standard substance, and the yield was 48.45% (Refer to the Appendix A for specific measurement and calculation methods).

The optical properties of prepared bCDs in water were investigated. As shown in Figure 2A, the yellowish–green line of bCDs displays strong absorption bands at 240 and 335, and moderate peaks between 450 and 700 nm. For the first peak at 240 nm, it was due to the π–π* transition in the sp^2^ domain [44]. The weak absorption peak shown at 335 nm corresponds to N−π* transitions of C=O or C=N bonds [45]. The last wide and weak peak was caused by the surface states induced by N atoms. The purple and blue lines in Figure 3A are the excitation wavelength (The optimal excitation wavelength was at 356 nm) and emission wavelength (The optimal emission wavelength was at 439 nm) curves of bCDs aqueous solution, respectively. Interestingly, bCDs appear yellowish–green under natural light, while bright blue fluorescence under the irradiation of 365 nm UV lamp. In addition, bCDs present an excitation wavelength-independent characteristic (Figure 2B,C). In detail, the emission peak of bCDs aqueous solution was always at 439 nm although the excitation wavelength increases from 290 nm to 420 nm.

### 3.2. Synthesis and Characterization of Td

For improving the efficiency of an aptamer to recognize *S. aureus*, a Td with an edge length of 18 bp was taken as the base, and a “handle” from one of the tips was used to hybrid with aptamer to identify the target. Amino/Desthiobiotin−modified Td was prepared by the method reported [40]. The Td showed clear bands on 3% agarose gel electrophoresis (AGE). As illustrated in Figure 3A, the weight of the product band increased with the increase in the nucleic acid chain, the strips of the five lanes (T1, T1 + T2, T1 + T2 + T3, Td, and Td1) appear from bottom to top. AFM was used to verify the assembly of Td; as shown in Figure 3B, Td was uniform in size of about 9.15 nm.

### 3.3. Synthesis and Characterization of Td1−MBs

Td1−MBs were prepared based on SA−desthiobiotin high−affinity coupling. The successful synthesis of Td1−MBs was verified by zeta potential experiments (Figure 4A). The measurement demonstrated that the average zeta potential of the SA−MB decreased from (20.77 ± 0.84) mV to (17.87 ± 2.41) mV after being modified by Td−des. The change of zeta potential resulted from the desthiobiotin of Td attaching to the SA on the surface of MBs. To achieve the highest combination efficiency, we subsequently explored the optimal amount of SA−MBs in a certain amount of Td−des. In Figure 4B, the absorption peak at 260 nm gradually weakened and approached 0, which means that with the increase in SA−MB usage, the remaining free nucleic acid in the supernatant decreased gradually to be saturated. When the volume of SA−MB exceeded 200 μL, Td−des in the solution was almost exhausted, indicating that 200 μL was the optimal usage of SA−MBs.

### 3.4. Synthesis and Characterization of Td2−bCD

The amidation of Td and bCD opens the possibility of using this Td2−bCD complex as a fluorescence signal amplifier. Ideally, each Td2 would combine with three bCDs to form a single signal amplifier monomer, but the bCD has amino groups to reckon with in addition to the considerable carboxyl group. BCDs can be aminated between each other, or between a bCD and multiple Td2s, to form unexpected complexes. After the proportion adjustment and optimization of the two, we attained uniform Td2−bCDs (Appendix A), and when the ratio of the two was 1:4 (Molar ratio of Td2: bCD), Td2−bCD reached the highest yield. Figure 5 demonstrates the shape of the signal amplifier observed under an AFM; most of the products were what we expected. Each of the three vertices of the tetrahedron had a bCD, which was evenly dispersed in the background at the height of about 15.5 nm (Figure 5B). These results meant the successful preparation of Td2−bCD, which lays a foundation for the smooth conduct of the experiment.

### 3.5. Sandwich Detection of S. aureus with bCD Fluorescence Signal Amplification

#### 3.5.1. Feasibility Verification and Signal Amplification Confirmation

In order to determine the feasibility of the detection strategy, *S. aureus* in HEPES buffer of 3.76 × 10^7^ CFU/mL was detected. The same operation was performed in the HEPES buffer without a target as a blank control. As expected, a strong fluorescence signal was detected (red line) in contrast with blank control (green line) (Figure 6). Furthermore, experiment signal amplification was verified by replacing Td1−MBs and Td2−bCDs with Apt1−MBs and Apt2−bCDs during the detection process. The yellow line in Figure 6 was the normalized luminescent intensity detected by Td1−MBs and Apt2−bCDs. By calculation, Td2−bCDs amplified the signal by 2.49 times. The blue line was the fluorescence intensity detected by naked Apt1−MBs and bCDs, and the signal amplification efficiency of the tetrahedral structure on the MB surface was 1.89 times (As opposed to the yellow line). Therefore, the signal amplification efficiency of the whole established detection method for *S. aureus* was 4.72 times.

#### 3.5.2. Detection of the Concentration of *S. aureus* and Specificity Test

Under suitable conditions, different concentrations of *S. aureus* were detected by this established detection platform, and the fluorescence signals obtained were plotted as Figure 7A,B, which displayed the fluorescence intensity of Td1@*S. aureus*@Td2−bCDs increased gradually with the increase in *S. aureus* concentration. As shown in Figure 7B inset, the fluorescence intensity has a strong linear relationship in the range of 7.22–1.44 × 10^9^ CFU/mL *S. aureus* (R^2^ = 0.9946), which was fitted the function: y = 178.99ln(x)−83.60. Additionally, the detection limit of the sandwich fluorescence biosensor for *S. aureus* was 4 CFU/mL, indicating that it has good sensitivity.

Results of *E. coli* O157:H7, *MRSA*, *V. parahaemolyticus*, *S. Typhimurium*, *S. aureus*, and a mixture of all the above bacteria at the same concentration were plotted in Figure 7C, for the sandwich detection platform, the influence of most bacteria was almost negligible, and only *MRSA* and *V. Parahaemolyticus* have a low response to the biosensor. This result demonstrated that the selectivity of this proposed method was satisfying.

#### 3.5.3. Detection Assays in Practical Samples

Sandwich detection of *S. aureus* with bCD fluorescence signal amplification has considerable selectivity, sensitivity, and anti-interference. To this end, broth, pasteurized milk, and drinking water were detected by this fluorescent probe. Table 1 shows the recovery rates of the strategy. With adding a concentration of 3.18 × 10^3^ CFU/mL *S. aureus* into real samples, the recovery rates were 104.72%, 102.20%, and 96.54%, respectively. Similarly, when adding 3.18 × 10^5^ CFU/mL *S. aureus*, the recovery rates were 97.80%, 103.14%, and 100.31%, respectively. Further, when adding a concentration of 3.18 × 10^6^ CFU/mL *S. aureus,* the recovery rates were 98.43%, 101.26%, and 99.37%, respectively. The above results showed that the method established in this work had satisfactory recovery rates and stability, and the reliability of the test in real food samples was confirmed.

## 4. Conclusions

In summary, a sandwich detection platform of *S. aureus* with bCD fluorescence signal amplification was innovated. Results showed that the fluorescence biosensor could be used as a sensitive probe for pathogen detection. The prepared bCD had high fluorescence quantum yield of 48.45%, and surface hydrophilic and rich groups make it easy to be modified by biological functionalization. With the advantage of the prominent enrichment ability of functionalized MBs, the Td−bCD fluorescence signal amplification system realized the sensitive detection of *S. aureus* in 2 h and amplified the detection signal by 4.72 times. The biosensor had acceptable stability with a recovery rate of 96.54–104.72% in real samples, which further demonstrated the potential application value of the biosensor. Moreover, the method can simply be transferred to other hazard detection by replacing aptamers. Nevertheless, the signal amplification element in this experiment has limited amplification efficiency; the more efficient signal-amplification strategies are still promising, such as aggregating more bCDs and Tds into larger microspheres.

## Data Availability

Not applicable.

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
