# Peer review of "Sandwich Fluorescence Detection of Foodborne Pathogen *Staphylococcus aureus* with CD Fluorescence Signal Amplification in Food Samples"

_foods, 2022, doi:10.3390/foods11070945_

Round 1

Reviewer 1 Report

The present paper shows development of Td-bCD (DNA tetrahedral-blue carbon quantum dot) fluorescence signal amplification system based biosensor for rapid, sensitive, and selective detection of S. aureus. The paper can draw greater attention to the readers if the authors revise the manuscript based on the following points:

  1. The developed method is based on sandwich detection of S. aureus using CD fluorescence signal amplification. The method comprises of a protocol composed of multiple steps. Followed by the usual culture and growth of S. aureus colony, bCD and two types of Tds (Td1 and Td2) have been prepared. Td1 is desthiobiotin functionalized DNA tetrahedral whereas Td2 is amine functionalized DNA tetrahedral. Thereafter, streptavidin-functionalized magnetic beads (MBs) have been attached to desthiobiotin functionalized DNA tetrahedral (Td1) to produce Td1-MBs. Subsequently, Td2-bCD has been prepared via reaction between -COOH of bCD surface with NH2 of Td2. In the next step, during the sandwich detection of S. aureus, Td1-MBs interacts with S. aureus by means of desthiobiotin functionalities. Thereafter, S. aureus-Td1-MBs have been treated with Td2-bCD to produce assembly of S. aureus-Td1-MBs and Td2-bCD. Finally, MBs were detached and separated from the assembly by treating the assembly with biotin. Finally, the assembly in the supernatant comprises of S. aureus-Td1 and Td2-bCD. The aforementioned process looks promising but it has been represented in a complicated manner. Accordingly, authors should represent it in a more simplified way.
  2. In the introduction part, some of very recently published papers, i.e., 10.1007/s12161-020-01821-4 and 10.1016/j.molliq.2017.06.106, should be referred.
  3. In the introduction, the strategy behind application of magnetic beads should be added.
  4. The composition of LB and the pH of HEPES buffer should be mentioned.
  5. In figure 1, the fluorescence maxima should be mentioned.
  6. The fluorescence intensity for S. aureus is much higher than Methicillin-resistant Staphylococcus aureus (Fig. 7c). What can be the reason(s)?

As a whole, the manuscript seems to get published after minor modifications required for simplification of sandwich detection process.

Author Response

Reviewer #1:

The present paper shows development of Td-bCD (DNA tetrahedral-blue carbon quantum dot) fluorescence signal amplification system based biosensor for rapid, sensitive, and selective detection of S. aureus. The paper can draw greater attention to the readers if the authors revise the manuscript based on the following points:

  1. The developed method is based on sandwich detection of aureus using CD fluorescence signal amplification. The method comprises of a protocol composed of multiple steps. Followed by the usual culture and growth of S. aureus colony, bCD and two types of Tds (Td1 and Td2) have been prepared. Td1 is desthiobiotin functionalized DNA tetrahedral whereas Td2 is amine functionalized DNA tetrahedral. Thereafter, streptavidin-functionalized magnetic beads (MBs) have been attached to desthiobiotin functionalized DNA tetrahedral (Td1) to produce Td1-MBs. Subsequently, Td2-bCD has been prepared via reaction between -COOH of bCD surface with NH2 of Td2. In the next step, during the sandwich detection of S. aureus, Td1-MBs interacts with S. aureus by means of desthiobiotin functionalities. Thereafter, S. aureus-Td1-MBs have been treated with Td2-bCD to produce assembly of S. aureus-Td1-MBs and Td2-bCD. Finally, MBs were detached and separated from the assembly by treating the assembly with biotin. Finally, the assembly in the supernatant comprises of S. aureus-Td1 and Td2-bCD. The aforementioned process looks promising but it has been represented in a complicated manner. Accordingly, authors should represent it in a more simplified way.

Response: Thank you for your valuable suggestion. Hope the revision is suitable for acceptance. The re-edited sentences were marked in blue.

Line 128-162:

2.3. Synthesis of CD

The synthesis of CD was adapted from previous reports [37]. 0.399 g of urea and 1.065 g of citric acid were dissolved in 20 mL of ultrapure water and mixed evenly. The mixed liquid was transferred to a high-pressure reactor for water and heated to 160℃ for the hydration reaction. After 6 h, the reaction liquid was cooled to room tempera-ture. The resulting solution was dialyzed in a 3500 Da dialysis bag for 4 hours and the solution outside the dialysis bag was collected. Finally, these solutions were dialyzed in a 500 Da dialysis bag to remove unreacted small molecules and to obtain pure blue CD (bCD). The prepared bCD was stored at 4℃ away from light for later use. The detailed characterizations were described in the Supporting Information.

2.4. Preparation of Td

In this work, we designed two Td constructures (Td1 and Td2) to couple with SA-MB and bCD respectively. To prepare Td with high yield, the authors referred to and modified the previous report [38,39]. The four single strands of Td (T1, T2-des, T3-des, and T4-des) were mixed with the same concentration (4 μM) and volume. Af-ter the chain disconnecting process at 95℃ for 10min, the mixture was cooled to 4℃ for at least 30 min. After that, the high yield Td1 could be obtained by mixing Apt1 (4 μM) with the above solution in a ratio of 1:4 by volume and standing. The Td2 solution was prepared in the same way, except that Apt1, T2-des, T3-des, and T4-des in the above operation was replaced by Apt2, T2-NH2, T3-NH2, and T4-NH2. The detailed characterizations and ratio optimization of raw materials were described in the Sup-porting Information.

2.5. Preparation of Td1 modified MBs (Td1-MBs)

Td1-MBs were produced by the strong interaction between desthiobiotin and SA. Suck up 300 μL of SA-MB and rinse it three times with HEPES buffer. 100 μL of Td1 and HEPES re-suspended SA-MB were evenly mixed and gently overturned and oscil-lated at 37℃ for 30min. After magnetic separation, the supernatant was discarded and the precipitate was suspended in HEPES again, leading to the formation of Td1-MBs.

2.6.  Preparation of Td2-bCD

Firstly, 2 µL bCD and 100 µL Td2 were mixed evenly, followed by freshly config-ured 7.5 mL of 10mg/mL EDC and 10.13 mL of 10mg/mL NHS. When activated by EDC, abundant -COOH on bCD surface will react with NH2 modified on Td2 to form semi-stable NHS, the semi-stable NHS will further react with a -NH2 to form an amide link. After sufficient vortices, the mixed solution was kept at room temperature over-night, away from light.

  1. In the introduction part, some of very recently published papers, i.e., 10.1007/s12161-020-01821-4 and 10.1016/j.molliq.2017.06.106, should be referred.

Response: Thank you for your valuable suggestion. We have added the two references in manuscript, marked in blue.

Line 66-68: CDs are relatively easy to synthesize and possess excellent biocompatibility [14], fa-vorable chemical stability [15], and high photostability [16,17].

Line 87-89: The binding specificity of aptamer and target is high and stable, and each target can bind thousands of aptamers on its surface [30].

  1. In the introduction, the strategy behind application of magnetic beads should be added.

Response: Thank you for your valuable suggestion. We have added the strategy behind application of magnetic beads in the introduction section. The re-edited sentences were marked in blue.

Line 81-86: Magnetic materials remain stable in complex non-magnetic samples and can chemically modify functional molecules on their surfaces, expanding their applications [22,23]. Previous studies have combined Fe3O4-based magnetic composite materials with fluorescence or colorimetry to easily separate and enrich targets using external magnetic fields [24,25], further improving the sensitivity of optical biosensor, suitable for the detection of trace substances in complex samples [26,27].

  1. The composition of LB and the pH of HEPES buffer should be mentioned.

Response: Thank you for your valuable suggestion. We have mentioned the pH of LB and HEPES buffer in manuscript.

Line 105-107: The Luria-Bertani culture (LB, pH=7.4) was bought from Sinopharm (Beijing, China), HEPES (pH=7.4), NaCl, 1-ethyl-3-(3-dimethylaminopropyl) carbodiimide hydrochlo-ride (EDC) was bought from Sigma-Aldrich LLC.

  1. In figure 1, the fluorescence maxima should be mentioned.

Response: Thank you for your valuable suggestion. The fluorescence properties was characterized in Figure 2, and we have re-edited the figure legend of Figure 2, the edited figure legend was marked in blue.

Line 243-247: Figure 2. Optical characterization results of bCDs. (A) UV/Vis absorption and fluo-rescence spectra at optimum excitation wavelengths (Inset: photo of bCDs solution in daylight and under 365 nm lamp); (B) Results of 3D-Scanning fluorescence emission spectra of bCDs at excitation wavelengths ranging from 280 to 420 nm; (C) Fluorescence emission spectra collected from the bCDs.

  1. The fluorescence intensity for aureus is much higher than Methicillin-resistant Staphylococcus aureus (Fig. 7c). What can be the reason(s)?

Response: In this study, the aptamer of Staphylococcus aureus was derived from the sequence screened in previous studies, which had strong specificity to Staphylococcus aureus (ATCC 6538) used in the experiment, but did not have strong specificity to recognize other strains of Staphylococcus aureus. If it is necessary to recognize all Staphylococcus aureus, aptamer sequences with this characteristic can be selected to meet the requirements.

As a whole, the manuscript seems to get published after minor modifications required for simplification of sandwich detection process

The manuscript has been revised as suggested and it is hoped that the current revised version will be accepted.

Reviewer 2 Report

Overall this is very interesting and novel work..  it should be published.  please consider revising the conclusion.. lines 325 -355.  this should be a conclusion NOT summary....include limitation and possible future direction...

Author Response

Reviewer #2:

Overall this is very interesting and novel work.. it should be published. please consider revising the conclusion. lines 325 -355. this should be a conclusion NOT summary....include limitation and possible future direction...

Response: Thank you for your valuable suggestion. We have re-edited the conclusion section (Marked in blue).

Line 343-356: In summary, a sandwich detection platform of S. aureus with bCD fluorescence signal amplification was innovated. Results showed that the fluorescence biosensor can be used as a sensitive probe for pathogen detection. The prepared bCD had high fluorescence quantum yield of 48.45% and surface hydrophilic and rich groups make it easy to be modified by biological functionalization. With the advantage of the prominent enrichment ability of functionalized MBs, the Td-bCD fluorescence signal amplification system realized the sensitive detection of S. aureus in 2 hours and amplified the detection signal by 4.72 times. The biosensor had acceptable stability with a recovery rate of 96.54%-104.72% in real samples, which further demonstrated the potential ap-plication value of the biosensor. Moreover, the method can simply be transferred to other hazard detection by replacing aptamers. Nevertheless, the signal amplification element in this experiment has limited amplification efficiency, the more efficient signal-amplification strategies are still promising, such as aggregating more bCDs and Tds into larger microspheres.

Reviewer 3 Report

Du et al. has carried out exciting work. The method and results are well elaborated and discussed. However, the manuscript needs critical editing and proofreading. I suggest authors either take help from a colleague who is an expert in academic writing or ask avail editing services from MDPI.

Abstract section does not give proper information. Abstract means a full-fledged summary that should give readers highlights of the information and experiments covered in the manuscript. Please revise the abstract (needs to rephrase and rewrite some sentences). Also, highlight essentialities and future perspectives of the study.

Section Introduction

The authors are advised to cut short the introduction and confine it to what is concerned to the contents of the manuscript. Besides, deepen it with why Staphylococcus aureus was tested.

Many parts are poorly written and very generalized. For example, but not limited to following sentences.

- May be present in samples of food contact surfaces 29 associated with their hosts. Besides, it is the work of a moment for S. aureus to spread 30 between the same or different species, and this wily bacteria has escaped from its hosts 31 (poultry, livestock, seafood) and can rapidly spread in the environment

- The worrying thing is that S. aureus starts out in small numbers, infects food or humans and quickly proliferate to disease-causing numbers.

- The selection of fluorescent signal molecules is very important for detection. Security and biocompatibility of signal molecules are critical when the analysts are food and creature.

- Results showed that the fluorescence biosensor can be used as a sensitive probe for pathogen detection.

Materials and Methods Section

-Take 1 mL bacteria liquid to centrifuge for 8min at 5000 rpm and wash with HEPES buffer 3 times, and bacteria at a concentration of CFU/mL were obtained by continuous 10-fold gradient dilution with sterile HEPES buffer.

-Line 123-124

- Suck up a certain amount of SA-MB and rinse it three times with HEPES buffer.

- When activated by EDC, abundant -COOH on bCD surface will react with NH2 modified on Td2. After sufficient vortices, the mixed solution was kept at room temperature overnight, away from light.

Scheme 1: Schematic diagram > both words have same meaning. Revise it to > Schematic of experimental principle.

Figure 5. Elaborate figure legend.

Section conclusion

The section needs to be more elaborative and should highlight the importance of the study and future directions with possible limitations.

-The prepared bCD had high fluorescence quantum yield and was easy to be modified by biological functionalization.

Author Response

Reviewer #3:

Du et al. has carried out exciting work. The method and results are well elaborated and discussed. However, the manuscript needs critical editing and proofreading. I suggest authors either take help from a colleague who is an expert in academic writing or ask avail editing services from MDPI.

Abstract section does not give proper information. Abstract means a full-fledged summary that should give readers highlights of the information and experiments covered in the manuscript. Please revise the abstract (needs to rephrase and rewrite some sentences). Also, highlight essentialities and future perspectives of the study.

Response: Thank you for your valuable suggestion. We have addressed all requested revisions carefully. Hope the revision is suitable for acceptance.

Line 11-25: Timely detection of Staphylococcus aureus (S. aureus) is critical because it can multiply to disease-causing levels in a matter of hours. Herein, a simply and sensitive DNA tetrahedral (Td) fluorescence signal amplifier with blue carbon quantum dots (bCDs) was prepared for sandwich detection of S. aureus. bCD was modified at the apex of Td, and aptamer on Td was used to accurately identify and "adsorb" the amplifier to the surface of S. aureus. Atomic force microscopy (AFM) demonstrates the successful preparation of this signal amplifier. The fluorescence intensity emitted in this strategy increased 4.72 times. The strategy showed a stronger fluorescence intensity change, sensitivity (linear range of 7.22-1.44 × 109 CFU/mL with a LOD of 4 CFU/mL), and selectivity. The recovery rate in qualified pasteurized milk and drinking water samples was 96.54% to 104.72%. Compared with simple aptamer sandwich detection, these fluorescence signal amplifiers have improved fluorescence detection of S. aureus. Additionally, this fluorescent signal amplification strategy may be applied to the detection of other food pathogens or environmental microorganisms in the future.

Section Introduction

The authors are advised to cut short the introduction and confine it to what is concerned to the contents of the manuscript. Besides, deepen it with why Staphylococcus aureus was tested.

Many parts are poorly written and very generalized. For example, but not limited to following sentences.

- May be present in samples of food contact surfaces associated with their hosts. Besides, it is the work of a moment for S. aureus to spread between the same or different species, and this wily bacteria has escaped from its hosts 31 (poultry, livestock, seafood) and can rapidly spread in the environment

- The worrying thing is that S. aureus starts out in small numbers, infects food or humans and quickly proliferate to disease-causing numbers.

Response: Thank you for your valuable suggestion. We have re-edited the sentences and marked in blue.

Line 37-43: S. aureus may be present in samples of food contact surfaces associated with their hosts, such as refrigerators, food bags, even shelves [4]. Moreover, S. aureus usually starts out in small numbers, infects food or humans and quickly proliferate to disease-causing numbers in 8-10 h. Therefore, detection of S. aureus contamination in time is crucial.

- The selection of fluorescent signal molecules is very important for detection. Security and biocompatibility of signal molecules are critical when the analysts are food and creature.

Response: Thank you for your valuable suggestion. We have re-edited the sentences and marked in blue.

Line 59-64: The selection of fluorescent signal molecules is very important for detection. When analyzing food and living things, the safety and biocompatibility of fluorescent signal molecules have a significant impact on whether new contaminants will be introduced into the analysis object [11].

- Results showed that the fluorescence biosensor can be used as a sensitive probe for pathogen detection.

Response: Thank you for your valuable suggestion. We have deleted the sentence.

Materials and Methods Section

-Take 1 mL bacteria liquid to centrifuge for 8min at 5000 rpm and wash with HEPES buffer 3 times, and bacteria at a concentration of CFU/mL were obtained by continuous 10-fold gradient dilution with sterile HEPES buffer.

Response: Thank you for your valuable suggestion. We have re-edited the sentences and marked in blue.

Line 124-127: Take 1 mL bacteria liquid to centrifuge for 8min at 5000 rpm and wash with equal volume HEPES buffer 3 times, and bacteria at a concentration of 101-108 CFU/mL were obtained by continuous 10-fold gradient dilution with sterile HEPES buffer.

-Line 123-124

Response: Thank you for your valuable suggestion. We have re-edited the sentences and marked in blue.

Line 146-147: After that, the high yield Td1 could be obtained by mixing Apt1 (4 μM) with the above solution in a ratio of 1:4 by volume and standing.

- Suck up a certain amount of SA-MB and rinse it three times with HEPES buffer.

Response: Thank you for your valuable suggestion. We have re-edited the sentence and marked in blue.

Line 153: Suck up 300 μL of SA-MB and rinse it three times with HEPES buffer.

- When activated by EDC, abundant -COOH on bCD surface will react with NH2 modified on Td2. After sufficient vortices, the mixed solution was kept at room temperature overnight, away from light.

Response: Thank you for your valuable suggestion. We have re-edited the sentence and marked in blue.

Line 161-162: When activated by EDC, abundant -COOH on bCD surface will react with -NH2 modified on Td2 to form semi-stable NHS, the semi-stable NHS will further react with a -NH2 to form an amide link.

Scheme 1: Schematic diagram > both words have same meaning. Revise it to > Schematic of experimental principle.

Response: Thank you for your valuable suggestion. We have changed the Scheme legend.

Line 179: Scheme 1. Schematic of experimental principle.

Figure 5. Elaborate figure legend.

Response: Thank you for your valuable suggestion. We have elaborated figure legend

Line 291: Figure 5. AFM result of Td-bCD. (A) Topview AFM images of Td-bCD, the field of view decreases from left (5 μm) to right (1 μm); (B) Td-bCD size distribution of the third image in (A).

Section conclusion

The section needs to be more elaborative and should highlight the importance of the study and future directions with possible limitations.

-The prepared bCD had high fluorescence quantum yield and was easy to be modified by biological functionalization.

Response: Thank you for your valuable suggestion. We have elaborated conclusion section

Line 346-358: In summary, a sandwich detection platform of S. aureus with bCD fluorescence signal amplification was innovated. Results showed that the fluorescence biosensor can be used as a sensitive probe for pathogen detection. The prepared bCD had high fluorescence quantum yield of 48.45% and surface hydrophilic and rich groups make it easy to be modified by biological functionalization. With the advantage of the prominent enrichment ability of functionalized MBs, the Td-bCD fluorescence signal amplification system realized the sensitive detection of S. aureus in 2 hours and amplified the detection signal by 4.72 times. The biosensor had acceptable stability with a recovery rate of 96.54%-104.72% in real samples, which further demonstrated the potential application value of the biosensor. Moreover, the method can simply be transferred to other hazard detection by replacing aptamers. Nevertheless, the signal amplification element in this experiment has limited amplification efficiency, the more efficient signal-amplification strategies are still promising, such as aggregating more bCDs and Tds into larger microspheres.

Round 2

Reviewer 3 Report

The authors have revised the manuscript. The manuscript can be accepted after the correction of typos. 

Herein, a simply and sensitive DNA > I think it should be " ... do you mean " simple